# Feedback-controlled active brownian colloids with space-dependent rotational dynamics

Miguel Angel Fernandez-Rodriguez [1,4], Fabio Grillo [1,4✉], Laura Alvarez [1], Marco Rathlef[1], Ivo Buttinoni [1,2], Giovanni Volpe [3] & Lucio Isa [1✉]

The non-thermal nature of self-propelling colloids offers new insights into non-equilibrium physics. The central mathematical model to describe their trajectories is active Brownian motion, where a particle moves with a constant speed, while randomly changing direction due to rotational diffusion. While several feedback strategies exist to achieve position-dependent velocity, the possibility of spatial and temporal control over rotational diffusion, which is inherently dictated by thermal fluctuations, remains untapped. Here, we decouple rotational diffusion from thermal fluctuations. Using external magnetic fields and discrete-time feedback loops, we tune the rotational diffusivity of active colloids above and below its thermal value at will and explore a rich range of phenomena including anomalous diffusion, directed transport, and localization. These findings add a new dimension to the control of active matter, with implications for a broad range of disciplines, from optimal transport to smart materials.

[1] Laboratory for Soft Materials and Interfaces, Department of Materials, ETH Zurich, 8093 Zurich, Switzerland. [2] Institut für Experimentelle Kolloidphysik, Heinrich-Heine University, 40225 Düsseldorf, Germany. [3] Department of Physics, University of Gothenburg, 41296 Gothenburg, Sweden. [4]These authors contributed equally: Miguel Angel Fernandez-Rodriguez, Fabio Grillo. ✉email: fabio.grillo@mat.ethz.ch; lucio.isa@mat.ethz.ch

The behavior of self-propelling colloidal particles sheds light on far-from-equilibrium physics and offers tantalizing opportunities to perform tasks beyond the reach of other micro- and nanoscale systems[1]. Many of these functions are inspired by the striking similarity that synthetic active matter exhibits with living systems such as motile bacteria. This analogy therefore also provides an ideal opportunity to understand the motion and (self-)organization of living systems through synthetic models[2]. The fundamental mathematical description of how active colloids move is given by active Brownian motion[1]: a microscopic particle of radius $R$ in a fluid of viscosity $\eta$ moves with constant speed $v$, while its orientation diffuses at a rate set by the rotational diffusivity $D_R = (k_B T)/(8\pi\eta R^3)$, with $k_B T$ being the characteristic thermal energy at absolute temperature $T$ and $k_B$ the Boltzmann constant. This minimal model has been successfully employed to describe a wealth of phenomena, from the motion of active particles in complex structures[3] to the optimization of search strategies[4].

Recently, there has been a growing interest in pushing the control of synthetic active matter beyond the standard active Brownian particle (ABP) model to mimic more complex behaviors, including directed transport and pattern formation. These phenomena typically arise when the particle velocity or the environmental fluctuations vary in space and time. For example, a position-dependent translational diffusivity has been proposed as a fundamental biological mechanism leading to anomalous diffusion and localization of biomolecules in cellular membranes[5], while the temporal control of random walks enables the emergence of collective motion in active colloids[6]. The effect of a position-dependent velocity has also been investigated as a means to control the organization and the area explored by active particles (artificial and biological) as well as their interactions[7–16]. Beyond their fundamental relevance, these mechanisms can also be exploited for applications ranging from environmental remediation to targeted drug delivery[1].

Because translation and rotation in ABPs are coupled, introducing a feedback between rotational dynamics and position also provides a means to control active Brownian motion. Biological swimmers, such as chemotactic bacteria[17], are in fact known to tune their rotational dynamics to climb up or down chemical gradients in order to localize food sources or to escape harmful chemicals. However, while biological swimmers can do so by varying their reorientation frequency (or tumbling rate)[18], which is an internal degree of freedom, the rotational dynamics of a synthetic active particle is usually dictated by thermal fluctuations.

Here, we control the rotational diffusivity of individual ABPs by decoupling the amplitude of the rotational fluctuations from the thermal bath. Through randomly-oriented magnetic fields and a discrete-time feedback loop, we spatially and temporally modulate the effective rotational temperature, above and below the environmental temperature. This allows us to study the effect of a position-dependent rotational diffusivity on the statistics of active Brownian motion. In analogy with biological and artificial sensor-actuator systems that rely on temporal sampling[17,19], we also consider that the feedback between $D_R$ and the particle's position is not instantaneous, but mediated by a discrete sampling of position that results in a finite sensorial delay. We find that periodic space-time modulations of the rotational dynamics bring about a broad range of exotic phenomena, ranging from anomalous diffusion reminiscent of glassy dynamics to directed transport and localization. We support our results with numerical simulations, which also indicate new directions for future developments.

## Results

**Controlling rotational dynamics.** Our model ABPs self-propel due to induced-charge electro-phoresis[20–22]. They consist of 4 µm-diameter silica Janus colloids, half-coated with a 120 nm-thick nickel cap, which is magnetized in the direction perpendicular to the Janus boundary (see inset in Fig. 1a). In this way, the propulsion's direction dictated by the compositional asymmetry is aligned with the caps' magnetic moment and can be externally controlled by a magnetic field. We let the particles sediment at the bottom of a liquid cell enclosed by two planar transparent electrodes, which are separated by a vertical gap $h = 120$ µm, and record the colloids' position and cap orientation at a frame rate of 10 fps by video microscopy (see Methods for more details). The colloids swim over the bottom substrate due to locally unbalanced electrohydrodynamic flows generated by a spatially uniform 1 kHz AC electric field applied across the electrodes (Fig. 1a). The swimming velocity $v$ is proportional to the square of the electric field $\propto (V_{pp}/h)^2$, where $V_{pp}$ is the peak-to-peak voltage, which is varied in the range 1–10 V.

We control the orientation angle $\theta$ of the colloids' cap (see microscopy image in Fig. 1a) by two pairs of independent Helmholtz coils generating spatially uniform magnetic fields of any in-plane orientation (Fig. 1a, Supplementary Fig. 1, and Supplementary Movie 1). In contrast to previous works, where magnetic fields are used to remote-control active colloids[23–27], we randomize the direction of the magnetic field to endow the colloids with an externally controlled rotational diffusivity, which is decoupled from the thermal bath and the propulsion scheme (Supplementary Movie 2). We vary the orientation of the magnetic field at $f = 1$ kHz by random angular displacements $\Delta\theta$ drawn from a Gaussian distribution with zero mean and variance $\sigma^2 = 2D_R/f$, where $D_R$ is the imposed rotational diffusivity. Fig. 1b shows the orientation angle of the magnetic field as a function of time for values of imposed $D_R$ ranging from $10^{-2}$ to 10 rad$^2$ s$^{-1}$. The corresponding distributions of the particles' angular displacements $G(\Delta\theta, \Delta t)$, measured from the cap orientation, attest that $\theta$ and the direction of the magnetic field diffuse according to the same Gaussian process (Fig. 1c). By letting $\theta$ diffuse over the entire $2\pi$ range, we can therefore enforce effective values of $D_R$ that are above and below the thermal rotational diffusivity $D_R^{th}$ ($1.4 \times 10^{-2}$ rad$^2$ s$^{-1}$ at room temperature). In particular, we can achieve rotational dynamics that are orders of magnitude faster than what the thermal bath would otherwise dictate (rotational cooling is also shown in the SI for 2 µm-diameter colloids in Supplementary Figs. 2 and 3).

External, independent control on $D_R$ and $v$ enables us to adjust the persistence of particle trajectories in real-time. For example, as demonstrated in Fig. 1d, we can gradually increase the propensity of an ABP to move along straight paths by decreasing $D_R$ over time in a step-wise fashion from 10 to $10^{-1}$ rad$^2$ s$^{-1}$, while keeping $v$ constant (Supplementary Movie 3, Supplementary Fig. 4 and Supplementary Movie 4 show the complementary case in which $D_R$ is kept constant and $v$ is varied in a step-wise fashion. See also Supplementary Supplementary Fig. 2 and Supplementary Fig. 3 for 2 µm colloids). An analysis of the mean squared displacements (MSD), calculated for each segment at a fixed $D_R$ (Fig. 1e), shows that the timescale at which the ABP's motion goes from being ballistic to being diffusive is proportional to $\sim D_R^{-1}$, as expected. As a result, the MSD at long times is larger for lower values of $D_R$, suggesting the possibility of controlling the area explored by an ABP by varying $D_R$ on demand. By systematically varying the imposed $D_R$ while fixing the AC voltage, and thus $v$, we can extract both $D_R$ and $v$ from the MSD to calculate the persistence length as $L_P = v D_R^{-1}$. Fig. 1f and its

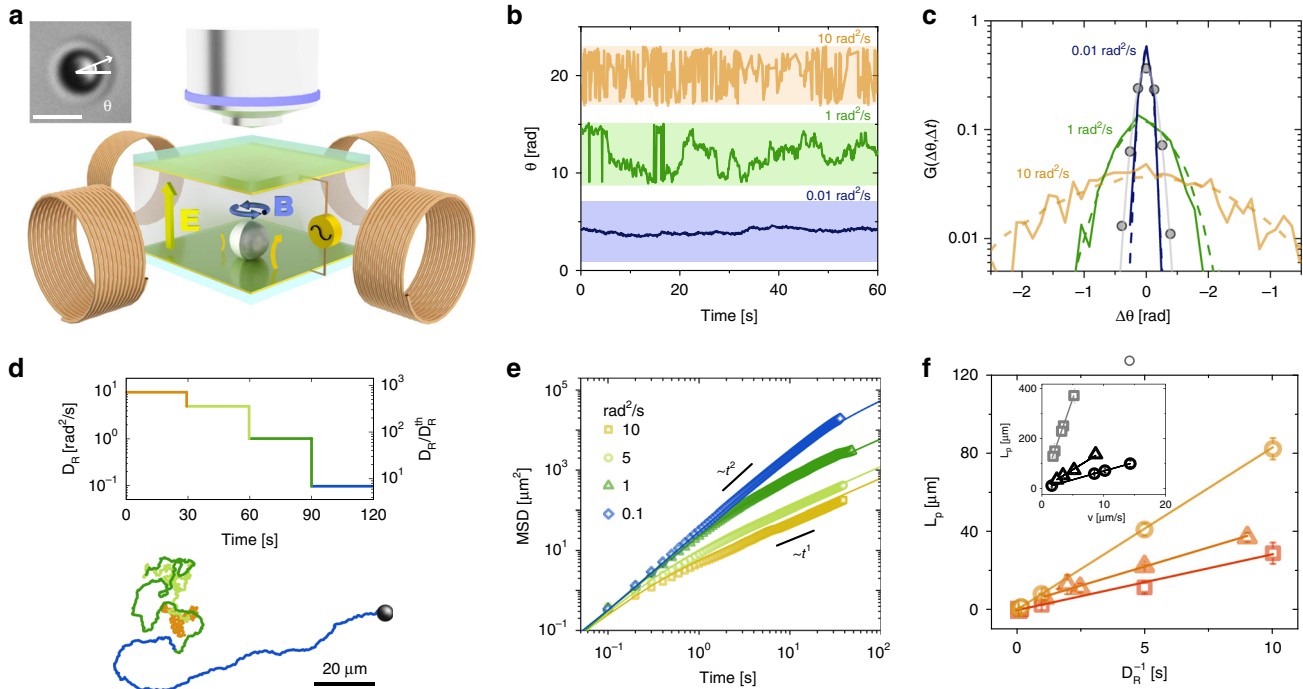

**Fig. 1 Controlling rotational dynamics through randomly-oriented magnetic fields. a** Schematic of the experimental setup. A Janus silica particle, half-coated with a magnetized Ni cap, undergoes self-propulsion at the bottom of a liquid cell enclosed by two transparent electrodes. Propulsion stems from induced-charge electro-phoresis generated by an AC electric field $\vec{E}$ perpendicular to the plane in which the particle moves (the curved yellow arrows depict local unbalanced electrohydrodynamic flows). A uniform magnetic field $\vec{B}$ (blue arrow), produced by four coils, controls the in-plane orientation of the particle, while its motion is observed with an optical microscope. The inset shows an optical micrograph of a 4 μm particle, with the Ni cap in black. The white arrow shows the particle's orientation angle $\theta$, which is aligned with the cap's magnetic moment and thus with the magnetic field. Scale bar: 5 μm. **b** Imposed $\theta$ as a function of time for three values of $D_R$. The colored bands delimit a $2\pi$ range. Data are shifted along the y-axis for clarity. **c** Probability distributions of angular displacements $G(\Delta\theta, \Delta t)$ for different $D_R$ and lag time $\Delta t = 0.1$ s (dashed lines: imposed $\Delta\theta$, solid lines: measured $\Delta\theta$) with the same colors as in **b**. The gray symbols show the measured $G(\Delta\theta, \Delta t)$ without magnetic field, corresponding to the thermal $D_R^{th}$ (gray line: Gaussian fit, see Supplementary Table 1). **d** Trajectory of an ABP with $v = 5.5$ μm s$^{-1}$ and an imposed $D_R$ varying over time. **e** MSD of an ABP with $v = 5.5$ μm s$^{-1}$ for different imposed $D_R$: 0.1 (diamonds), 1 (triangles), 5 (circles) and 10 (squares) rad$^2$ s$^{-1}$ (colors as in **d**). **f** Persistence length ($L_P$) as a function of rotational relaxation time $D_R^{-1}$ for different values of $v$: 8.2 (circles), 6.7 (triangles), and 2.7 (squares) μm s$^{-1}$. The inset shows $L_P$ as a function of $v$ for the thermal $D_R^{th} = 0.014$ rad$^2$ s$^{-1}$ (gray squares) and for an imposed $D_R$ of 0.07 and 0.144 rad$^2$ s$^{-1}$ (black triangles and circles, respectively). Error bars correspond to the data standard deviation.

inset show that $L_P$ can be varied over a wide range of values, displaying the expected linear scaling on both control parameters. These data thus attest to our ability to engineer the motion of ABPs by independently controlling $v$ and $D_R$ using electric and magnetic fields.

**A feedback loop for position-dependent rotational dynamics.** To study the impact of space-time modulations of the rotational dynamics on the statistics of active Brownian motion, we implemented a discrete-time feedback loop that updates $D_R$ based on the ABP's position $\mathbf{r}(t)$ (Fig. 2a, Methods, and Supplementary Figs. 5 and 6). Similarly to the case of the non-instantaneous response of motile microorganisms to environmental cues[17], we update $D_R$ at regular intervals based on the past ABP's position $\mathbf{r}(t - \tau)$, where $t = n\tau$, $\tau$ is the sampling period and $n$ is the number of samples. This is realized in the experiments by holding $D_R$ constant between consecutive sampling periods, and in the Langevin dynamics simulations by letting the rotational friction vary according to a zero-order hold (ZOH) model[19], as described in the Methods section (see Fig. 2a). Moreover, in analogy with Brownian motion in periodic potentials, which is a paradigmatic model for the description of anomalous diffusion[28–31], here we let $D_R$ vary according to a checkerboard pattern of alternating square regions of size $L$. In each square, $D_R$ takes on either a high ($D_R^H$) or a low ($D_R^L$) value (Fig. 2b). Specifically, we consider scenarios

where $D_R^H = 10$ rad$^2$ s$^{-1}$, $D_R^L = 0.01$ rad$^2$ s$^{-1}$, and $L/v$ such that $\frac{1}{D_R^L} > \frac{L}{v} >> \frac{1}{D_R^H}$ (Fig. 2c, d and Supplementary Movie 5 and Supplementary Fig. 7). These choices imply that the motion is predominantly diffusive in one region and ballistic in the other, with two well-separated relaxation timescales.

Given the existence of multiple parameters that affect the dynamics and the spatial organization of the ABPs, in the following subsections we examine them, and highlight their main contributions, separately. We begin by determining the role of the characteristic timescale $L/v$ and continue with the effect of the sampling period $\tau$, before concluding with an analysis on particle localization.

**Non-Gaussianity: L/v and exponential tails.** We find that our feedback scheme coupled with modulations of $D_R$ brings about different types of anomalous diffusion at different time and length scales, depending on the sampling period $\tau$ and the timescale $L/v$.

We begin by examining the statistics of particle motion for nonzero but small sampling periods $\tau << L/v$. In this regime, while the trajectories within each region are either ballistic or diffusive (Fig. 2c–d), the overall dynamics presents unique features. The distribution of one-dimensional displacements, rescaled by $L$, $G(|x|/L, \Delta t)$ (Fig. 3a–d) presents different types of non-Gaussianity at different lag times $\Delta t$ depending on the ratio $\Delta t(L/v)^{-1}$. We

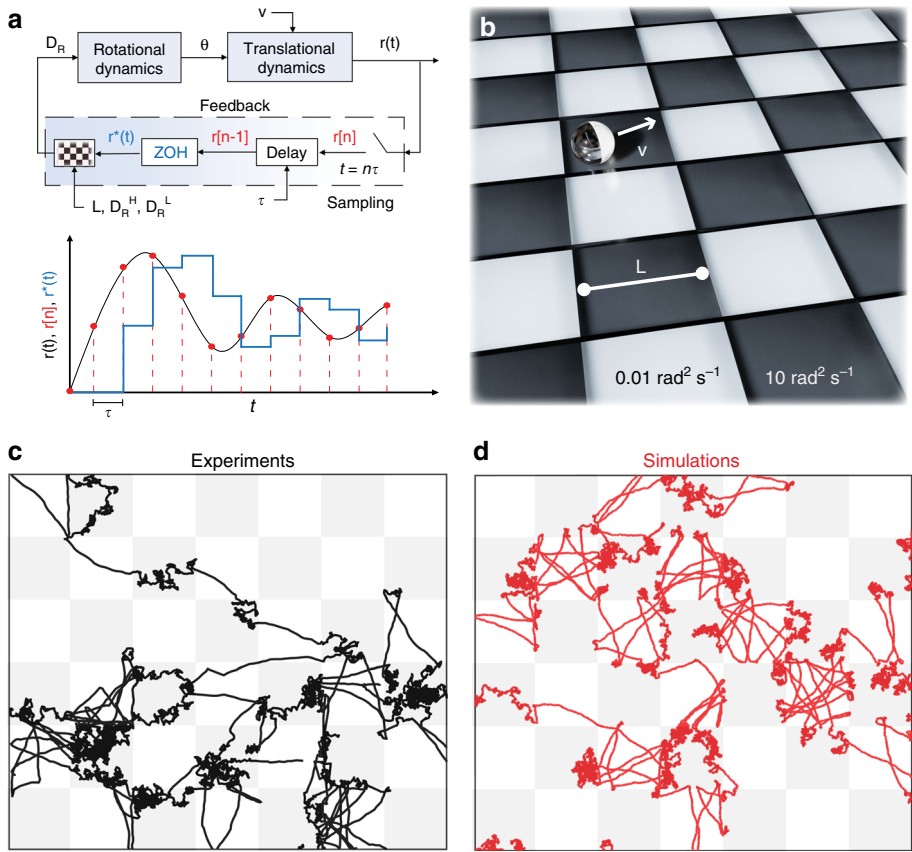

**Fig. 2 Position-dependent rotational dynamics: sampling a checkerboard pattern with a sensorial delay. a** (top) Block diagram illustrating the feedback loop enforced both in experiments and simulations to achieve space-time modulations of the rotational dynamics. The feedback function that couples $D_R$ with the particle's position r(t) consists of four blocks in series: sampling, delay, zero-order hold (ZOH), and $D_R = f(r)$. **a** (bottom) Schematic representation of how r(t) is sampled at discrete time points $t = n\tau$ (corresponding to r[n]) and reconstructed via a ZOH model (r*(t)) after a delay of $\tau$, i.e., using the position r[n-1] as input. **b** Schematic of an ABP moving over a checkerboard pattern of rotational diffusivity. **c–d** Side-by-side comparison of experimental (black) and simulated (red) trajectories of ABPs with $D_R$ varying according to the checkerboard pattern. ($v = 3.5$ μm s$^{-1}$, $L = 32$ μm, $D_R^H = 10$ rad$^2$ s$^{-1}$, $D_R^L = 0.01$ rad$^2$ s$^{-1}$, and $\tau = 0.4$ s).

quantify the departure from Gaussian behavior in terms of the excess kurtosis (Fig. 3e): $\gamma = \frac{\langle x^4 \rangle}{\langle x^2 \rangle^2} - 3$, where $\gamma = 0$ for a normal distribution. For $\Delta t << L/v$, $G(|x|/L, \Delta t)$ is broader and flatter than a normal distribution ($\gamma < 0$), which is consistent with the broadening of $G(x, \Delta t)$ reported for ABPs at intermediate timescales, when the motion is dominated by ballistic segments[32]. However, as $\Delta t$ approaches $L/v$, $G(|x|/L, \Delta t)$ develops into a leptokurtic distribution ($\gamma > 0$) characterized by a Gaussian peak at small $|x|$ followed by an exponential tail up to $|x| \simeq L$, after which it rapidly decays.

Exponential tails in $G(x, \Delta t)$ have already been observed in glassy systems and for the diffusion of colloids in macromolecular environments[33–35]. They are often explained in terms of dynamic heterogeneity, which is the coexistence of faster and slower particles, which explains their appearance also in our system. At timescales $\Delta t \simeq L/v$, we also have two distinct populations of particles, which travel over very different length scales depending on where they reside on the checkerboard: diffusive in $D_R^L$-regions and ballistic in $D_R^H$-regions. Because $\Delta t \simeq L/v >> D_R^H$, ABPs in $D_R^H$-regions are effectively diffusive, with a $D_{eff} \simeq v^2/D_R^H$ [1], and thus their displacements are normally distributed. They are responsible for the Gaussian peak, and travel distances much smaller than $L$. Conversely, ABPs in $D_R^L$-regions move in a ballistic fashion because $\Delta t \simeq L/v << 1/D_R^L$, and thus their displacements scale as $v\Delta t$, which are of order $L$. Interestingly,

while dynamic heterogeneity in glassy colloidal systems arises from hindered translational diffusion due to steric interactions with other particles[36], here it is the result of a spatially heterogeneous rotational dynamics. As we will show later, the parallel with glassy systems goes even further.

As we examine dynamics at longer $\Delta t$, the excess kurtosis $\gamma$ attains a positive maximum at $\Delta t \simeq L/v$ (Fig. 3e and Fig. 4a–b) and later decays to zero, indicating that Gaussian statistics is eventually recovered at long times, as expected. Interestingly, the MSD exhibits a super-diffusive scaling $\sim\Delta t^{1.7}$ up to the same critical timescale, after which the diffusive regime ($\sim\Delta t^1$) is gradually recovered in the limit of long time scales (see Fig. 3f). This in stark contrast to the dynamics of ABPs with constant $D_R$. In the latter case, $\gamma$ is negative at intermediate timescales and zero on short and long timescales, and a transition from a ballistic (MSD $\sim t^2$) to a diffusive (MSD $\sim t^1$) scaling of the MSD is found at $\Delta t \simeq 1/D_R$[32].

Finally, at a given $\tau << L/v$, the timescale $L/v$ dictates not only the $\Delta t$ at which the maximum degree of non-Gaussianity is attained, but also its extent (Fig. 4a, b). This is again a direct consequence of the motion being the combination of distinct ballistic and diffusive segments. The excess kurtosis $\gamma$ grows with $\Delta t$ because the extent of the exponential tail grows faster with $\Delta t$ ($\sim\Delta t$) than the broadness of the Gaussian peak ($\sim\sqrt{\Delta t}$). Given that ABPs can travel in a ballistic fashion only up to $\sim L$, the tail of $G(|x|/L, \Delta t)$, and thus $\gamma$, keeps growing up to $\Delta t \sim L/v$. Hence,

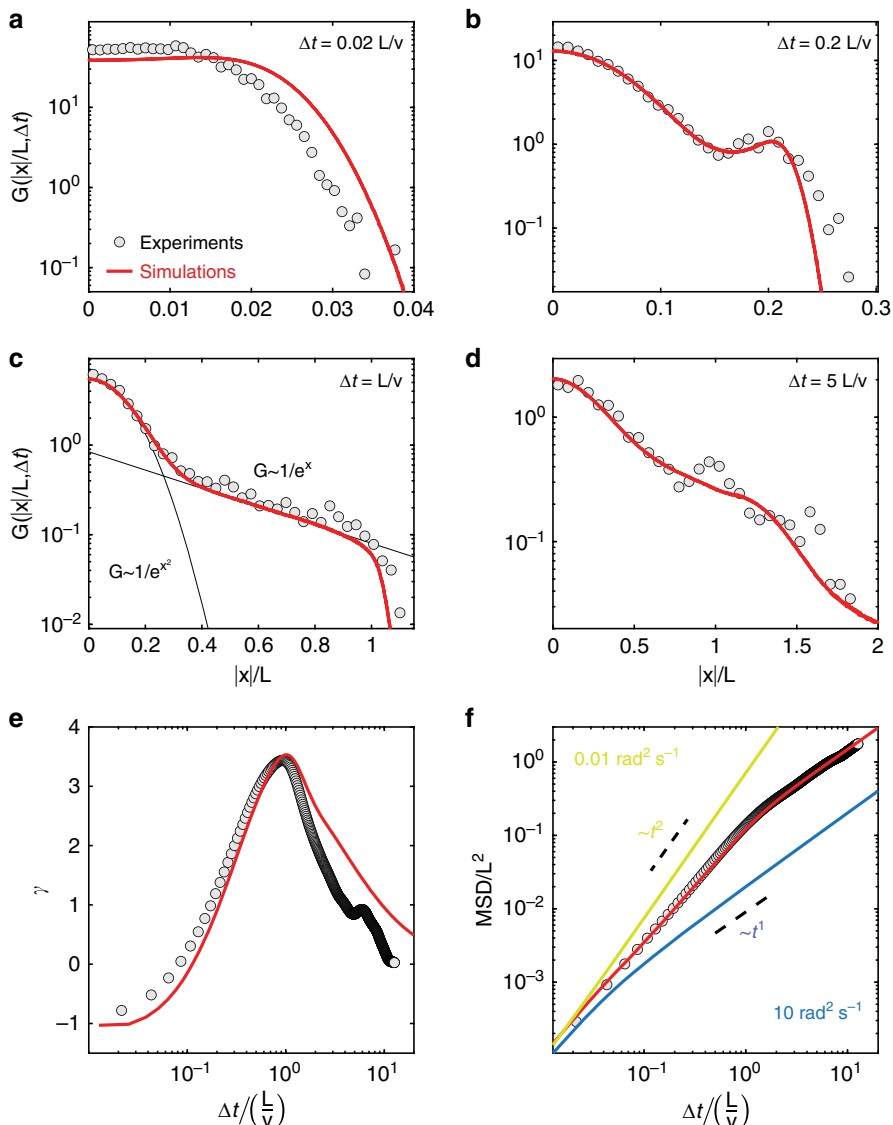

**Fig. 3 Non-Gaussian statistics and exponential tails due to $D_R$ varying according to a checkerboard pattern.** Experimental and simulated data are plotted as circles and red lines, respectively. **a–d** Probability distributions of rescaled one-dimensional displacements $G(|x|/L, \Delta t)$, for $\Delta t(L/v)^{-1} = 0.02$ (**a**), 0.2 (**b**), 1 (**c**), and 5 (**d**). One-dimensional displacements in x and y are cumulated to increase statistics. The black lines in the third panel ($\Delta t(L/v)^{-1} = 1$) are a Gaussian and an exponential fit to $G(|x|/L, \Delta t)$ in the range $|x|/L \simeq 0-0.2$ and $\simeq 0.4-1$, respectively. **e** Excess kurtosis $\gamma$ of $G(x/L, \Delta t)$ as a function of $\Delta t$ $(L/v)^{-1}$. **f** Rescaled mean squared displacement $MSD/L^2$ as a function of $\Delta t(L/v)^{-1}$. The yellow and blue straight lines are the theoretical MSDs of ABPs with constant $D_R$, equal to $D_R^L$ and $D_R^H$, respectively ($v = 4$ μm $s^{-1}$, $L = 32$ μm, $D_R^H = 10$ rad$^2$ $s^{-1}$, $D_R^L = 0.01$ rad$^2$ $s^{-1}$, and $\tau = 0.2$ s).

the greater $L/v$, the greater the maximum $\gamma$, which is correspondingly attained at larger $\Delta t$. It is worth noting that, in the limiting case of small $L/v \lesssim 1/D_R^H$, an exponential tail cannot develop and as a result $\gamma$ remains negative, approaching zero in the limit of long times, as in the case of ABPs with constant $D_R$ (Fig. 4a).

**Non-Gaussianity: $\tau$ and subdiffusion.** The sampling period $\tau$ has a qualitative impact on the statistics of motion depending on its value relative to the timescale $L/v$ (Fig. 4c–h). In particular, we identify three regimes: $\tau = 0$, which corresponds to an instantaneous update of $D_R$ based on the ABP's position, $\tau < L/v$ and $\tau > L/v$.

For $\tau = 0$ (Fig. 4c, black circles and Supplementary Fig. 8) $\gamma$ attains multiple local maxima at $\Delta t(L/v)^{-1} = 1$, 3, and 5, with the global maximum being at $\Delta t(L/v)^{-1} = 3$. Such maxima in $\gamma$ also mark changes in the scaling of the MSD (Fig. 4e), which exhibits a

superdiffusive scaling $\sim \Delta t^{1.7}$ up to $\Delta t \simeq L/v$, after which it starts to saturate and attain a subdiffusive scaling (MSD $\sim \Delta t^a$, $a < 1$) up to $\Delta t \simeq 3-5L/v$, with higher $L/v$ leading to smaller $a$. For $\Delta t \gg L/v$, the MSD eventually recovers the diffusive scaling ($a = 1$).

The emergence of subdiffusion reinforces the previously mentioned analogy with glassy dynamics. Nonetheless, in our system, there is no physical caging by neighboring particles[37]. Subdiffusion is instead the result of an "effective dynamical caging", arising from the randomization of the direction of motion of ballistic ABPs as these enter $D_R^H$-regions. In fact, up to $\Delta t \simeq L/v$, the MSD is dominated by ballistic segments of length $\sim v\Delta t$ in $D_R^L$-regions. Conversely, over the same timescale, diffusive ABPs in $D_R^H$-regions only travel comparatively negligible distances $\sim v\sqrt{\Delta t/D_R^H}$. However, over timescales $\gtrsim L/v$, ballistic ABPs cannot travel distances greater than $\simeq L$ without crossing into a $D_R^H$-region. As they do so, there is a finite probability that

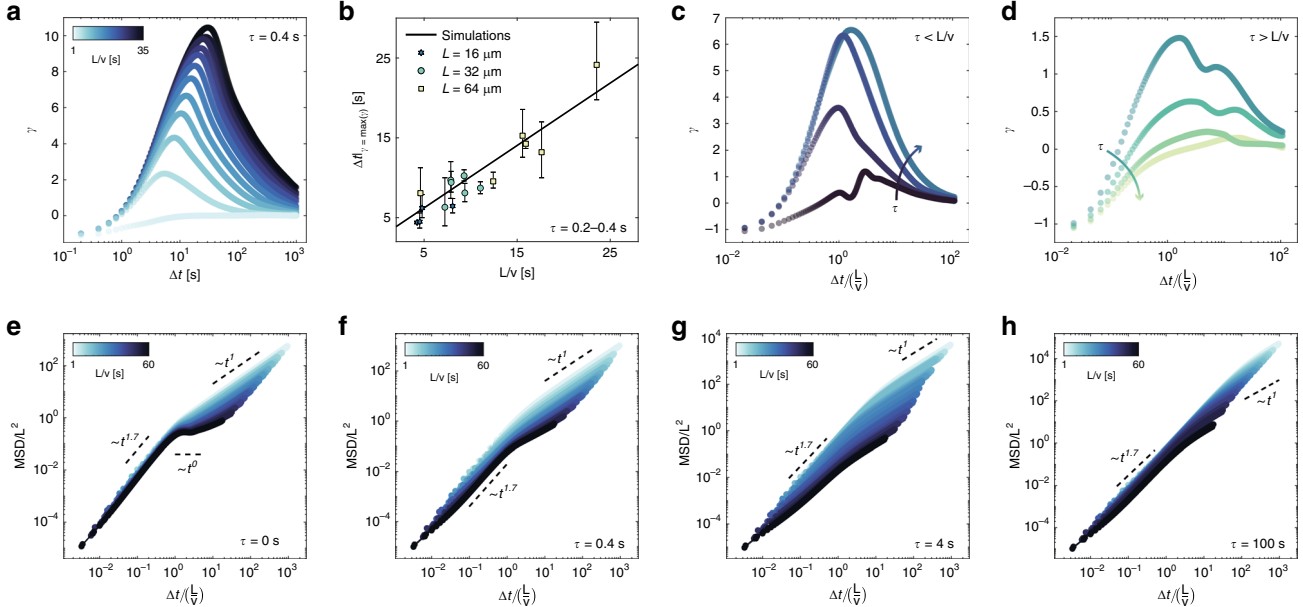

**Fig. 4 Role of the timescale $L/v$ and sampling period $\tau$ on the statistics of particle motion. a** Simulated evolution of the excess kurtosis $\gamma$ of the distribution of one-dimensional displacements $G(x/L, \Delta t)$ with $\Delta t$ for different values of $L/v$ at a constant $\tau = 0.4$ s. **b** Experimental and simulated $\Delta t$ at which $\gamma$ attains its maximum as a function of $L/v$ ($\tau = 0.2$–$0.4$ s). Error bars denote 95% confidence intervals. **c-d** Simulated evolution of $\gamma$ with $\Delta t(L/v)^{-1}$ for $\tau < L/v$ (**c**) and $\tau > L/v$ (**d**), with $L = 32$ μm and $v = 3.5$ μm s$^{-1}$. The black circles corresponds to $\tau = 0$. **e-h** Simulated evolution of the rescaled mean squared displacement MSD/$L^2$ with $\Delta t(L/v)^{-1}$ for different values of $L/v$ and $\tau$.

they diffuse back and cross the $D_R^L$-region from which they came in the opposite direction (Supplementary Movies 6 and 7). Such reflection events give virtually null displacements over timescales up to $\sim 2L/v$, the time ballistic ABPs take to cross a $D_R^L$-region and travel back. The MSD then grows only as $\simeq v^2/D_R^H \Delta t$ for those particles that cross a $D_R^L$-region an odd number of times and remain in a $D_R^H$-region. Finally, such a dynamical caging, and the subsequent subdiffusive motion, become less prominent with decreasing $L/v$, as the difference between diffusive and ballistic displacements diminishes (Fig. 4e).

Moving to nonzero values of $\tau$ brings about qualitative changes to the characteristics of $\gamma$ depending on whether $\tau < L/v$ or $\tau > L/v$. Up to $\tau \sim L/v$, higher values of $\tau$ translate into higher values of $\gamma$, into the increased prominence of the maximum at $\Delta t \simeq L/v$, and into the gradual disappearance of local minima (Fig. 4c). Because nonzero values of $\tau$ introduce a finite delay in the update of the rotational dynamics, ballistic ABPs can penetrate into $D_R^H$-regions up to lengths $\sim v(2\tau)$ before adjusting their rotational dynamics (see Fig. 2a). This not only increases the effective distance that ABPs can travel in a ballistic fashion, but can also allow them to cross entire regions without adjusting their rotational dynamics. The increase in the maximum value of $\gamma$ with increasing values of $\tau$ is therefore due to a larger disparity between the relative contributions of ballistic and diffusive displacements.

Increasing $\tau$ beyond $\simeq L/v$ leads to the gradual decrease of $\gamma$ across intermediate timescales and the disappearance of a maximum in the limit of $\tau \gg L/v$ (Fig. 4d). For $\tau > L/v$, the modulations of the rotational dynamics start to depart considerably from the inherent periodicity of the checkerboard pattern. In the limit of large $\tau$, the rotational dynamics of the ABPs is determined by their initial position rather than the region over which they move. This implies the existence, at all times, of two different populations of particles moving in an either ballistic or diffusive fashion, whose $D_R$ remains constant for a period of time equal to $\tau$. In this case, the overall dynamics is the mere

superposition of the dynamics of ABPs with different $D_R$. Therefore, in the limit of large $\tau$, $\gamma$ does not present a maximum and remains negative or close to zero (Fig. 4d).

Values of $\tau > 0$ also lead to the disappearance of subdiffusion at $\Delta t \simeq L/v$ (Fig. 4f–h). Nonetheless, the MSD still displays a crossover between a superdiffusive and a diffusive scaling on short and long timescales, respectively. The disappearance of subdiffusion is again due to the fact that a finite $\tau$ allows ballistic ABPs to penetrate into $D_R^H$-regions up to greater lengths before updating their $D_R$. This fact not only minimizes the probability that ABPs diffuse back into the $D_R^L$-region from which they entered, but also increases the chances that ballistic ABPs cross entire $D_R^H$-regions without updating their $D_R$, thus contributing to higher MSDs. For analogous reasons, increasing $\tau$ also causes the superdiffusive-to-diffusive transition to take place at $\Delta t > L/v$ because ballistic ABPs can travel distances greater than $L$.

**Localization**. While a position-dependent rotational dynamics alone cannot sustain pattern formation[38,39], the finite sensorial delay introduced by the sampling ($\tau > 0$) in the feedback loop leads to the localization of ABPs in $D_R^H$-regions (Fig. 5). Interestingly, the degree of localization is a nonmonotonic function of $\tau(L/v)^{-1}$. For instantaneous updates of $D_R$ ($\tau = 0$), the time-averaged steady-state spatial distribution $\rho(x, y)$ is homogeneous, with no manifestations of the underlying $D_R$ pattern (Fig. 5a, numerical simulations). As $\tau$ is increased, $\rho(x, y)$ increases in $D_R^H$-regions at the expense of $D_R^L$-regions up to a maximum degree for a critical $\tau \simeq 0.1$–$0.3 L/v$, before returning to a homogeneous distribution in the limit of large $\tau(L/v)^{-1}$ (Fig. 5a).

We further quantify the degree of departure from the homogeneous distribution by studying the evolution of the ratio between the simulated average $\rho(x, y)$ in $D_R^H$- and $D_R^L$-regions: $\rho^H$ and $\rho^L$, respectively (Fig. 5b, c). In agreement with the density maps shown in Fig. 5a, the $\rho^H/\rho^L$ ratio is 1 in the limit of small or large $\tau$ and presents a maximum at intermediate values of $\tau$

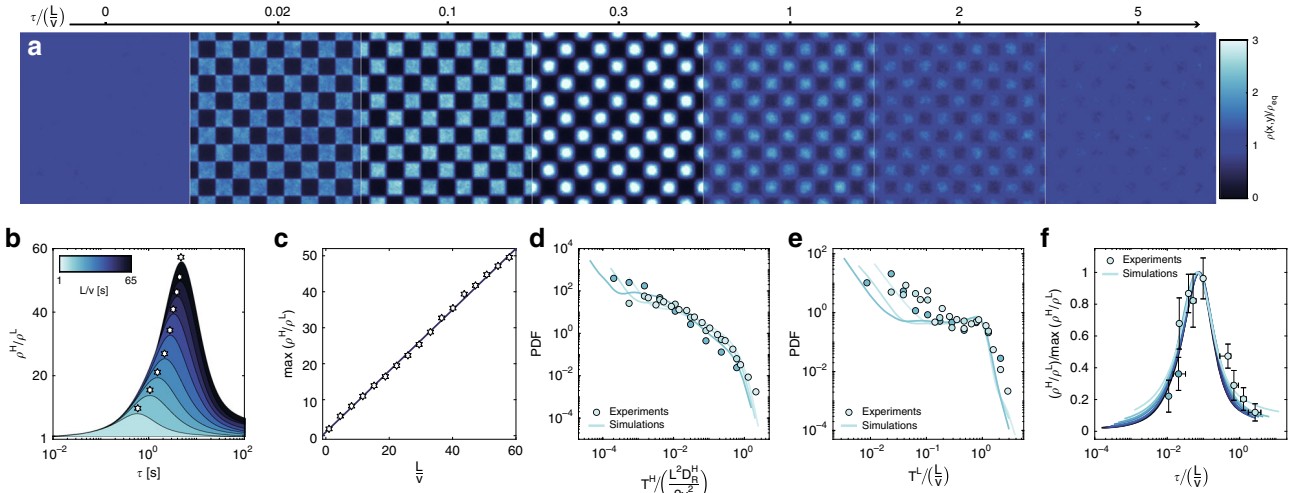

**Fig. 5 Emergence of localization for finite sampling period $\tau$. a** Simulated particle density distribution $\rho(x, y)$ for different values of $\tau(L/v)^{-1}$, rescaled by the uniform distribution $\rho_{eq} = 1/A$, where $A$ is the area of the simulation box. $\rho_{eq}$ is recovered for a spatially homogeneous $D_R$ or $\tau = 0$. The simulation parameters are $v = 3.5\ \mu m\ s^{-1}$, $L = 32\ \mu m$, $D_R^H = 10\ rad^2\ s^{-1}$ and $D_R^L = 0.01\ rad^2\ s^{-1}$. The distributions are obtained by binning the positions of $2.5 \times 10^4$ particles for a simulated time of 900 s, after letting the particles move from their initially random positions for 100 s. Periodic boundary conditions are enforced over a square simulation box of size $10L \times 10L$. **b** Simulated ratio between the average particle densities in the $D_R^H$- and $D_R^L$-regions, $\rho^H/\rho^L$, as a function of $\tau$ for different values of $L/v$. $\rho^H/\rho^L$ is calculated as $\langle T_H/T_L \rangle$, where $T_H$ and $T_L$ are the residence times in the $D_R^H$- and $D_R^L$-regions, respectively. **c** Simulated maximum $\rho^H/\rho^L$ (stars) as a function of $L/v$, fitted to a linear model (black line). **d–e** Simulated and experimental distribution of **d** $T_H$ and **e** $T_L$, normalized by the effective diffusive timescale $\frac{L^2 D_R^H}{2v^2}$ and the ballistic timescale $L/v$, respectively. **f** Experimental and simulated $\rho^H/\rho^L$ as a function of $\tau(L/v)^{-1}$ for different $L/v$, rescaled by the corresponding maximum. Lines and circles in **d–f** are color-coded based on $L/v$ according to colormap of **b**. Error bars denote 95% confidence intervals.

(Fig. 5b). In particular, the maximum $\rho^H/\rho^L$ ratio is a linearly increasing function of $L/v$ (Fig. 5c).

The existence of a maximum degree of localization and its dependence with $L/v$ can be explained with a simple transport argument based on the dynamic asymmetry introduced by finite values of $\tau$. As previously mentioned, ballistic ABPs can in fact penetrate into $D_R^H$-regions up to lengths $\simeq v(2\tau)$ before updating their $D_R$, whereas diffusive ABPs keep diffusing up to lengths $\sim v\sqrt{D_R^{-1}\tau}$ in between updates. The deeper ballistic ABPs can penetrate into diffusive regions, the longer it takes them to diffuse out. Therefore, nonzero values of $\tau$ imply that, overall, particles end up spending more time in $D_R^H$-regions. This picture is in qualitative agreement with the higher degree of localization in the center of the $D_R^H$-regions for $\tau \simeq 0.3L/v$ (Fig. 5a).

More quantitatively, because at steady state the net flux between regions must be zero, we can write:

$$\frac{\rho^H}{T^H} = \frac{\rho^L}{T^L}, \tag{1}$$

where $T^H$ and $T^L$ are the average residence times of the ABPs in $D_R^H$- and $D_R^L$-regions, respectively. Therefore, $max(\rho^H/\rho^L)$ is equal to $max(T^H/T^L)$. For $0 < \tau \ll L/v$, we expect $T^H$ and $T^L$ to scale as $\sim(L/v)^2$ and $\sim L/v$, respectively, because the residence time in a $D_R^H$-region depends on the time that an ABP takes to penetrate into it ($\sim L/v$) and to diffuse out of it ($\sim \frac{L^2}{2v^2/D_R^H}$). Hence, the maximum of the ratio $\rho^H/\rho^L = T^H/T^L$ scales linearly with $L/v$, as shown in Fig. 5c.

The different scaling of $T^H$ and $T^L$ is also confirmed by the experimental and simulated residence time distributions at different $L/v$ (Fig. 5d–e). In particular, the distributions of $T^H$ and $T^L$ collapse onto the same master curves, when rescaled by $\frac{L^2}{2v^2/D_R^H}$ and $L/v$, respectively, and quickly drop to zero for rescaled values of $T^H$ and $T^L$ greater than 1. Moreover, by renormalizing the $\rho^H/\rho^L$ ratio by its respective maximum for a given $L/v$, and

plotting it as a function of $\tau(L/v)^{-1}$, we find that both simulated and experimental data collapse onto a single master curve (Fig. 5f).

Finally, in the limit of $\tau > L/v$ the degree of localization starts to drop because at such large sampling periods the rotational dynamics is decoupled from the inherent periodicity of the underlying checkerboard pattern. This cross-over can be viewed in terms of the Nyquist–Shannon's theorem[19]. When $\tau \sim L/v$, the frequency at which the ABPs sample their environment is comparable to the highest frequency at which the ABPs can cross a region of the checkerboard pattern. This means that for $\tau > L/v$ the ABPs cannot sense changes of $D_R$ happening on a length scale $L$. At all times, the ABPs will retain a given $D_R$ for a period equal to the sampling period. This leads to two different populations moving in an either ballistic or diffusive fashion depending on their respective initial positions, which are updated every $\tau$.

Interestingly, for $\tau > L/v$, the disconnect between the ABP's sampling resolution and the spatial periodicity of the underlying pattern brings about a degree of localization that oscillates over time (Fig. 6). Nonetheless, such oscillations are damped for $\tau \gg L/v$ due to lingering correlations of particle motion based on past positions introduced by the delay component in the feedback loop. By removing the delay, while retaining the discrete sampling, such that $D_R$ is updated every $t = n\tau$ according to the ABP's current position $\mathbf{r}(t)$ rather than the past one $r(t - \tau)$, we can obtain instead persistent oscillations with a period equal to $\tau$ (Fig. 6a, b).

## Discussion

Our findings illustrate that engineering the feedback between the internal dynamics (e.g., $D_R$, $v$) and the state ($\mathbf{r}(t)$) of ABPs allows tailoring their response, both in terms of the statistics of motion at the single-particle level and in relation to their global spatio-temporal organization. In particular, we show that the response is defined by the balance of timescales in the system, the ones characteristic of active Brownian motion. i.e., set by $v$ and $D_R$, and

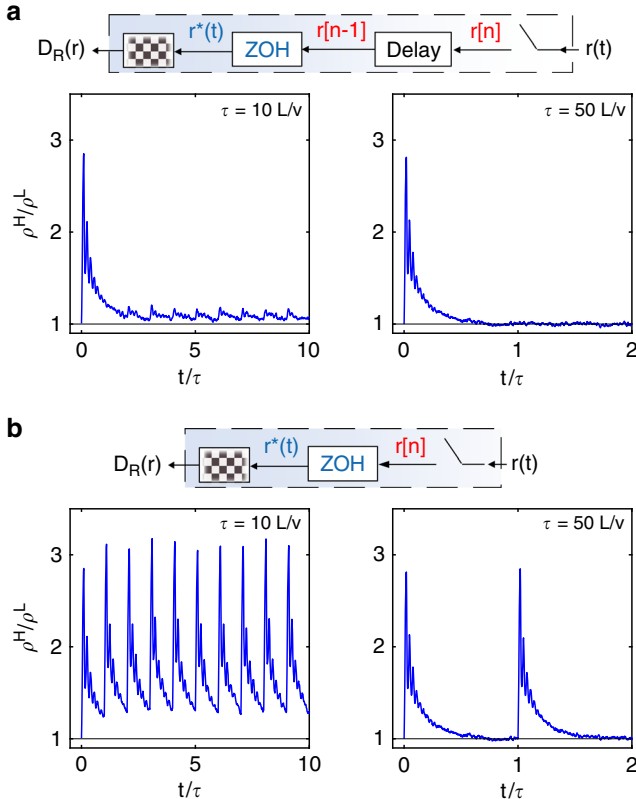

**Fig. 6 Oscillating localization for $\tau > L/v$ with and without delay in the feedback loop. a** Simulated time evolution of $\rho^H/\rho^L$, directly calculated from the density distribution $\rho(x, y)$ at each time step. $D_R$ is updated every $t = n\tau$ based on the past particle's position $\mathbf{r}(t - \tau)$. **b** Same as in **a** except for that $D_R$ is updated every $t = n\tau$ based on the current position $\mathbf{r}(t = n\tau)$. The simulation parameters are $v = 3.5\ \mu m\ s^{-1}$, $L = 32\ \mu m$, $D_R^H = 10\ rad^2\ s^{-1}$ and $D_R^L = 0.01\ rad^2\ s^{-1}$.

the externally imposed ones, set by the modulation of dynamical landscapes and the feedback clock. Our experiments show that, by decoupling rotational fluctuations from the thermal bath, we now have full control on each of these timescales, which enables us to begin exploring new directions, where numerical simulations play an essential role in providing guidelines and large statistics for the validation of the results.

Looking toward the future, our findings open up new interesting avenues to direct the dynamics and organization of ABPs. Local control over the rotational dynamics offers an alternative means to control the persistence of active trajectories at a given velocity, which can be harnessed to optimize the navigation of ABPs in complex environments[40,41]. The introduction of periodic modulations in the rotational dynamics of ABPs also defines a new framework to study a variety of anomalous diffusion phenomena[28–31], beyond the cases presented here, with the emergence of interesting analogies with glassy dynamics, as discussed above. Since ABPs enable directed transport and pattern formation, even in the absence of particle interactions and external forces[38,42,43], introducing various forms of feedback, communication[2], delay[44], information flows[11] and sensory ability[12] defines new opportunities. Borrowing ideas and tools from signal processing and control systems, we envisage the engineering of more complex dynamical responses. We could, for instance, adapt ideas developed for nuclear detectors (dead-time analysis) or robotic systems (feed-forward responses or negative delays[39,45]) to devise new signal reconstruction strategies between discrete sampling events for ABPs, or design higher-order or

integral responses to mimic the way in which biological microswimmers sense and adapt to chemical signals[46]. Finally, a last challenge, which also constitutes an exciting opportunity, is to translate the capabilities provided by external feedback and control into internal responses, e.g., through adaptation and reconfiguration[46], in order to develop fully autonomous artificial microswimmers.

## Methods

**Fabrication of Janus particles.** In order to fabricate our active Janus particles, silica colloids with 4.28 μm diameter (5% w/v, microParticles GmbH, Germany) are diluted to 1:6 in MilliQ water and spread on a glass slide, previously made hydrophilic by a 2-minute air plasma treatment. Upon drying of the suspension, a close-packed particle monolayer is formed. The monolayer is then sputter-coated with 120 nm of nickel (Safematic CCU-010, Switzerland) to create the Janus surface. After coating, the glass slide is placed overnight above a neodymium magnet ($50 \times 50 \times 12.5\ mm^3$, 1.2 T) to align the magnetic moments of all particles in the direction of the compositional asymmetry. The particles are retrieved from the glass slide by pipetting and withdrawing a droplet of water on top of the monolayer. An identical procedure is followed for 2 μm silica particles, for which data are shown in the SI.

**Cell preparation.** The transparent electrodes are fabricated from 24 mm × 24 mm No. 0 coverglasses (85–115 μm-thick, Menzel Gläser, Germany) covered with 3 nm of chromium and 10 nm of Au deposited by metal evaporation (Evatec BAK501 LL, Switzerland), followed by 10 nm of $SiO_2$, deposited by plasma enhanced chemical vapor deposition (STS Multiplex CVD, UK) to minimize the adhesion of particles to the substrate. A water droplet containing the particles is deposited on the bottom electrode within the 9 mm-circular opening of a 0.12 mm-thick sealing spacer (Grace Bio-Labs SecureSeal, USA). The top and bottom electrodes are connected to a signal generator (National Instruments Agilent 3352X, USA) that applies the AC electric field, with a fixed frequency of 1 kHz and varying the $V_{PP}$ voltage between 1 and 10 V. For 5 V, the applied electric field is 42 V mm⁻¹.

**Experiments.** The magnetic moment of the Janus particles is confined to the electrode plane and freely rotated within it through a custom-built setup fitted with two pairs of independent Helmholtz coils[47]. The magnetic field is constant within a few percent over a 1 mm² area in the center of the cell and the maximum applicable magnetic field is 65 mT. In the experiments, we use a field of 30–40 mT to orient the particles. In order to impose an effective rotational diffusivity to the particles, the magnetic field angle at step $n + 1$ ($\theta_{n+1}$) is obtained by adding to $\theta_n$ a random angular displacement $\Delta\theta$, which is given by Eq. (2), where $D_R$ is the target rotational diffusivity, $\Delta t$ is the time step (1 ms in our experiments), and $\eta$ is a random number sampled from a normal Gaussian distribution.

$$\theta_{t+1} = \theta_t + \sqrt{2D_R\Delta t}\eta \qquad (2)$$

The Janus particles are imaged with a home-built bright-field microscope in transmission and image sequences are taken with a sCMOS camera (Andor Zyla) at 10 fps with a $512 \times 512$ pixels² field of view. The image series to measure the thermal and imposed rotational diffusivity are acquired using a 50× objective (Thorlabs). The positions of the center of the JPs and of the metal cap are located using Labview routines. Then, a vector connecting both centers is used to determine the orientation of the particle at each frame for different $D_R$, from which the angular displacement distribution in Fig. 1c was calculated. The image series of ABPs actuated by the magnetic and the AC electric fields, are acquired with a 10× objective (Thorlabs). In this case, only the particle center of mass is located, and all the dynamical information is extracted from the final particle trajectory.

For the experiments with space-dependent $D_R$, single particles are located in real time by a custom LabView software. Series of $1024 \times 1024$ pixels² images are recorded at 5.88 fps. The coordinates are used to update the particle $D_R$ based on a predefined landscape. For the data presented in the main text, the field of view is divided into checkerboard patterns with $5 \times 5$, $10 \times 10$ and $20 \times 20$ squares, respectively, with alternating regions of $D_R^H = 10\ rad^2\ s^{-1}$ and $D_R^L = 0.01\ rad^2\ s^{-1}$. $D_R$ is updated every $\tau$, using values ranging from the smallest possible delay of 170 ms to $\tau = 17$ s, based on the particle coordinates at $t - \tau$. We vary the ballistic timescale $L/v$ by varying $L$ between 16 and 64 μm um and $v$ in the range 3–12 μm s⁻¹.

The particle thermal translational ($D_T^{th}$) and rotational ($D_R^{th}$) diffusion coefficients at room temperature (24 °C) are extracted from their 2D trajectories in the absence of magnetic and electric fields.

**Numerical simulations**. We simulate the dynamics of the ABPs by solving the equations of motion:

$$m\ddot{x} = f_x(\theta) - \gamma_T \dot{x} + \sqrt{2k_B T \gamma_T} \eta_x(t)$$
$$m\ddot{y} = f_y(\theta) - \gamma_T \dot{y} + \sqrt{2k_B T \gamma_T} \eta_y(t) \tag{3}$$
$$I\ddot{\theta} = \gamma_R(x, y, \tau)\dot{\theta} + \sqrt{2k_B T \gamma_R(x, y, \tau)} \eta_\theta(t),$$

where $m$ and $I$ are the mass and the moment of inertia of the colloid, respectively, $f_x(\theta)$ and $f_y(\theta)$ are the $x$ and $y$ components of the active force acting on the colloid, $\gamma_T$ is the friction coefficient associated with translational motion, $\gamma_R(\mathbf{r}, \tau)$ is the state-dependent friction coefficient associated with rotational motion, and $\eta_x(t)$, $\eta_y(t)$, and $\eta_\theta(t)$ are uncorrelated random numbers satisfying:

$$\langle \eta_x \rangle = \langle \eta_y \rangle = \langle \eta_\theta \rangle = 0; \ \langle \eta_x^2 \rangle = \langle \eta_y^2 \rangle = \langle \eta_\theta^2 \rangle = 1. \tag{4}$$

The active forces $f_x(\theta)$ and $f_y(\theta)$ are set equal to $\gamma_T v \cos(\theta)$ and $\gamma_T v \sin(\theta)$, respectively, such that in the absence of thermal noise and in the limit of long times the particles move at a constant velocity equal to $v$. We solved Eq. (3) in the underdamped limit through a Verlet-type integration scheme proposed by Gronbech-Jensen and Farago using the Itô convention[48]. Although Eq. (3) could also be solved in the overdamped limit, this approach allowed us to achieve a faster convergence to the homogeneous distribution for $\tau = 0$ using a relatively small integration step $dt = 0.001$ s.

For a position-dependent $D_R$, we let the rotational friction $\gamma_R(\mathbf{r}) = k_B T / D_R(\mathbf{r})$ vary as a function of the ABP's position $\mathbf{r} = [x(t), y(t)]$ according to a checkerboard pattern as follows:

$$\gamma_R(\mathbf{r}) = \frac{\gamma_R^H - \gamma_R^L}{2} \left\{ 1 + \text{sgn}\left[ \sin\left(\frac{\pi x}{L}\right) \sin\left(\frac{\pi y}{L}\right) \right] \right\} + \gamma_R^L, \tag{5}$$

where:

$$\text{sgn}(x) = \begin{cases} 1, & x \geq 0 \\ -1, & x < 0 \end{cases}, \tag{6}$$

and $\gamma_R^H$ and $\gamma_R^L$ correspond to the regions of high and low $D_R$, respectively.

For the implementation of the discrete time-feedback loop using the ZOH model, we update $\gamma_R$ every $t = n\tau$, where $n$ is the number of samples. Since the rotational diffusivity is a physical quantity that must be continuous in time, we reconstruct it from discrete-time inputs using the following function:

$$\gamma_R(\mathbf{r}, \tau) = \sum_{j=-\infty}^{+\infty} \gamma_R[j] \Pi(t - n\tau), \tag{7}$$

where $\gamma_R[j]$ is $\gamma_R(\mathbf{r})$ evaluated at $\mathbf{r}(t = j\tau)$:

$$\gamma_R[j] = \int_{-\infty}^{+\infty} \gamma_R(\mathbf{r}) \delta(t - j\tau) dt, \tag{8}$$

$\Pi$ is a rectangular function defined as:

$$\Pi = \begin{cases} 1, & 0 \leq t < \tau \\ 0, & \text{otherwise} \end{cases}. \tag{9}$$

and $j$ is an integer number which is set equal to $n - 1$ in the simulations for a delay equal to $\tau$, and to $n$ for the those without delay.

## Data availability

The data that support the findings of this study are available from the corresponding authors upon reasonable request.

## Code availability

The code used in this study is available from the corresponding authors upon reasonable request.

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

## Acknowledgements

We thank Hartmut Löwen for insightful discussions. L.I., M.A.F.R., and L.A.F. acknowledge financial support from the Swiss National Science Foundation Grant PP00P2-172913/1.

## Author contributions

Author contributions are defined based on the CRediT (Contributor Roles Taxonomy) and listed alphabetically. Conceptualization: L.A., I.B., M.A.F.R., F.G., L.I., G.V. Formal analysis: L.A., F.G. Funding acquisition: L.I. Investigation: L.A., M.A.F.R., F.G., L.I., M.R. Methodology: L.A., I.B., M.A.F.R., F.G., L.I. Project administration: L.I. Software: L.A., M.A.F.R., F.G., G.V. Supervision: L.I. Validation: L.A., M.A.F.R., F.G. Visualization: L.A., M.A.F.R., F.G., L.I. Writing—original draft: L.A., M.A.F.R., F.G., L.I. Writing—review and editing: L.A., I.B., M.A.F.R., F.G., L.I., G.V.

## Competing interests

The authors declare no competing interests.
