## [Peer Review File · Nature Communications]

Reviewers' comments:

Reviewer #1 (Remarks to the Author):

This manuscript presents an active matter system with a tunable rotational diffusion coefficient, an avenue exploited by several biological systems. The authors implement this with a clever magnetic control system, and show, with the aid of simulations, that this system has controllable diffusive behavior from sub-diffusive to ballistic. They also argue for localization appearing in the system, and present an interesting analogy to the Nyquist theorem to explain their results.

The authors clearly demonstrate fine control over rotational diffusion, and show that spatial modulation can lead to a variety of behavior, but I do not believe the results are sufficiently novel to warrant publication in this journal. Both aspects of the work (a swimmer with tunable D_R and spatial modulation of activity) have been explored recently (Karani et al., PRL 123 208002 (2019), Koumakis et al., Soft Matter 15 7026-7032 (2019), Arlt et al., Nature Communications 10 2321 (2019)), albeit separately. Additionally, the most compelling aspect of the work, the localization presented in Figure 4, is shown only in simulations, with no explanation for why this phenomena was (or was not) observed in the experiments. To add further confusion, in the section of the manuscript discussing figure 4 in is not at all clear that the authors have switched to discussing simulations.

Reviewer #2 (Remarks to the Author):

This manuscript investigates the dynamics of active Brownian particles whose rotational diffusivity depends on their position in the 2D-plane. The authors first present an experimental realisation of such a system, consisting of Janus particles that propel in the 2D plane when an AC electric field is applied. In addition, the coated hemisphere of the particles is made to possess a magnetic moment in the direction of propulsion, that aligns with an applied magnetic field. By imposing random variations of the latter combined to a feedback loop, one can alter the dynamics of rotational diffusion, above or below the equilibrium value imposed by the thermal bath. Their feedback loop allows to define a spatially dependent rotational diffusivity, here a chessboard of low and high values, that the particles sense with a given sampling time. They experimentally show that a rich non-Gaussian translational dynamic emerges, which markedly differs from that of active particles with a constant rotational diffusivity, but resembles that of supercooled liquids of interacting passive particles. Using simulations, they show an accumulation (resp. depletion) of particles in regions of high (resp. low) rotational diffusivity, and the sharpness of this pattern can be optimised by varying the sampling time.

I found the manuscript interesting and well organised. The authors demonstrate a nice control of rotational diffusivity above and below its equilibrium value. The experiments and simulations address an important paradigm in collective and transport phenomena, namely how the variation of motility and diffusivity of active particles induce pattern formation. The authors convincingly illustrate that spatial variations of the rotational diffusivity induce pattern formation only if individuals adapt to their environment with some finite delay. They show how non-Gaussian dynamics can emerge as a combination of space-dependent rotational diffusivity and constant self-propulsion, which is not obvious at first sight.

However, I find that the discussions related to Fig. 3 and 4 are unclear and need revision before publication (see my points below). I remain on the fence as for the relevance to a journal like Nature Communications due to the impact of the experimental system. Obtaining good statistics seems prohibitively long in the experiment so many aspects are studied via simulations. Because multiple particles in the field of view would have the same orientations and noise history, it seems to me that the experimental system does not allow for the study of collective effects, which is a drawback. That said, this experimental system may have potential as a probe to drive out of equilibrium various

complex materials in a controlled manner.

1. The experimental system is well characterised, and in particular, the control of the rotational diffusivity is well established: the measured distribution of angles corresponds to the imposed one, which testifies that no lag is present in the response of the particle to the field actuation. Could the authors include a short comment on the influence of the field's intensity to achieve such control though?

2. Fig. 2(e) it takes a very trained eye to evaluate the gaussian/non-gaussian character of the distributions. Would it be possible to plot these in y-log scale instead (gaussians will appear as parabolas and the non-gaussian tails will be clearly visible)? Also, I was wondering what those distributions look like for the rotational degree of freedom.

3. Fig. 2(f) the non-gaussian parameter does not fully decay to zero here but it does in the simulated data. Is it physical or simply due to a lack of statistic in the experiment? Or to the fact that the time lag does not go beyond 100s, i.e. does not exceed $1/D_R L^2$?

4. The first important result shown here is that non-interacting active Brownian particles with constant propulsion speed exhibit non-gaussian translational dynamics when their rotational diffusivity varies as a function of position. The dynamics resemble passive interacting particles in supercooled liquids but is markedly different from pure active Brownian particles. Nevertheless, I found this section very descriptive of the plots and not always clear. What I am missing is a statement that would summarise and conclude on the influence of the two parameters L/v and τ to help the reader. I would also like the following points to be addressed:

- Page 7, line 135: "For small L/v , the length scale of the spatial variation in D_R is smaller than any persistence length". I guess this regime is achievable in simulations, but not in experiments since the smallest square size is 16 μm , while the smallest persistence length is $v/D_R H = 3.5/10 = 0.35\mu\text{m}$. I think I need clarification on why the behaviour is that of a homogeneous rotational diffusivity here.

- In experiments, is L/v varied through L only or v as well?

- Page 8, line 153: " α reveals a super-diffusive scaling ..." The different dynamic regimes are easier to see from the MSD than the non-gaussian parameter. The mentioned cut-offs for the sub-diffusive plateau L/v and $3L/v$ seem loosely defined: the yellow band in Fig. 3(d) is clearly much wider than the actual plateau.

- Page 9, line 159: the peaks at odd multiples of L/v in Fig. 3(c) were not obvious to me on an x-log scale. In all honesty, I thought it was just a lack of statistics at first and would have missed this feature. Could they be magnified in an inset? What would the corresponding MSD look like?

- Page 9, line 160: "Consequently, the MSDs on timescales L/v decrease and the sub-diffusive scaling gradually vanishes (see Figure 3e). For $\tau > L/v$, α decreases again with no detectable peaks. In this regime, the MSDs increase across all timescales (Fig. 3f)." I am afraid I cannot see a decreasing MSD in Fig. 3(e) and the last statement is a bit empty, since all visible MSDs do increase.

5. The second important result is that pattern formation can be achieved with a spatially varying rotational diffusivity combined to a non-zero sampling time. Spatial variation of the rotational diffusivity alone does not lead to pattern formation, mainly because it is absent from the flux balance condition. As shown by the authors using simulations, the particle assembly is homogeneous at low

and high sampling time. At intermediate values though, particles are localized mainly in regions of high rotational density. The authors discuss the behaviour of the optimum localisation in terms of the sampling times and the crossing time L/v . I like that the authors propose a simple model to estimate the maximum particle fraction that can be localized in high rotational diffusivity regions. But I find the argument unclear and unnecessary long:

- Page 10, line 178: it should be specified that D_R here is D_R^H
- From line 178 still, the penetration time into a ballistic zone is τ but then another time, Δt^* , is introduced in line 185. Aren't they the same?
- The explanation of the model is a bit disorganised and gets heavy in the SI while getting Equation 3 only takes a few steps I find (see additional information in the attached document).

Other minor points:

- Page 2, line 50: "a change of tumbling rate can be straightforwardly mapped into an effective increase of rotational diffusivity". From Ref. 14 it is true that the mapping itself is straightforward but the conditions of its validity seem less so.
- Page 2, line 51: "nonetheless, translational and rotational fluctuations are coupled via the thermal bath and their lower bound is dictated by thermal fluctuations". This is only true for active Brownian particles but for run and tumble particles, the tumbling rate is an internal degree of freedom which is not coupled to the bath.
- Page 3: the magnetic moment is mentioned in line 65 but the external control of rotational diffusivity is only described much later in line 73. I think this is a bit confusing for the reader.
- Page 5, line 101: there is a typo in the word "instantaneous".
- Page 7, line 127: it may be worth specifying that this non-gaussian parameter is defined for 1D displacements. Why is there a factor 1/5 in its definition?
- Page 7, line 131: I think I would rephrase and say "zero on both short and long timescales".
- Page 7, line 139: γ_R is not introduced in the main text.
- Page 10, line 192: $\langle \eta_t \rangle$ is not introduced in the main text.

Reviewer #3 (Remarks to the Author):

In their paper the authors demonstrate a quite remarkable control of the rotational diffusion of an active Janus swimmer using a rotating magnetic field. This has never been achieved before and opens the road towards a full control of active swimmers.

The paper is well written, the data and methods are clearly presented and discussed.

And I only have a few minor points to address (see below)

I therefore recommend publication as soon as these minor points are solved.

Minor comments

Fig 2 (e-f) : indicate that the black points are experimental data and that the red lines are numerics

Fig 2 End of caption : one of the D_R^L must be replaced by D_R^H

Just before Formula 1 : replace At equilibrium by In the steady state

The description of the diffusive regimes is a bit lengthy and lacks some explanations, which only come in the Conclusion and Discussion part. Some reorganization of the writing could easily fix that.

It would be nice to synthesize the overall findings in a diagram as a function of $\tau L^{-1} v$. A comment about the role of $v/(D_R H L^2)$ would also be welcome.

Response to Reviewers' comments:

We thank the Reviewers for their comments, which we have taken into consideration in the revision of our manuscript. In order to address their comments, we have extensively revised the manuscript performing new experiments, analysis and numerical simulations. In particular:

- We have clarified the role of the delay τ in the feedback scheme for the modulation of D_R in the text and in a new version of Figure 2a. We have correspondingly carried out a complete new set of numerical simulations with the updated feedback scheme.
- We have performed a new analysis of the experimental displacement distributions as a function of lag time, which is now reported in the new Figure 3. This allowed us to present an additional discussion of the analogies and differences with glassy dynamics, now more clearly discussed in the text.
- We have included new extensive experimental data in the new Figure 4b, showing that the scaling of the position of maximum of the non-Gaussian parameter extends to more combinations of L and v for small values of τ .
- We have performed a broad range of new experiments to study the degree of localization for many combinations of τ and L/v , before and after the optimum τ for localization (new Figure 5f). The new data show excellent agreement with the numerical scaling.
- We have performed a new analysis of the distribution of residence times in the regions of high and low rotational diffusivities and compared them to the numerical data (new Figure 5d-e).
- The analysis in terms of residence time distributions allowed us to propose a simple rescaling argument for the degree of localization, now described in Figure 5b-f.
- We have compared two different feedback schemes, with and without sensorial delay, in the limit of large τ , revealing the presence and the absence of oscillations in the spatial organization of our ABPs (new Figure 6), expanding on the previous version of the manuscript.

In addition to these major modifications, we respond to the detailed comments of each reviewer below. We report their original comments in italics.

Reviewers' comments:

Reviewer #1 (Remarks to the Author):

This manuscript presents an active matter system with a tunable rotational diffusion coefficient, an avenue exploited by several biological systems. The authors implement this with a clever magnetic control system, and show, with the aid of simulations, that this system has controllable diffusive behavior from sub-diffusive to ballistic. They also argue for localization appearing in the system, and present an interesting analogy to the Nyquist theorem to explain their results.

The authors clearly demonstrate fine control over rotational diffusion, and show that spatial modulation can lead to a variety of behavior, but I do not believe the results are sufficiently novel to warrant publication in this journal. Both aspects of the work (a swimmer with tunable D_R and spatial modulation of activity) have been explored recently (Karani et al., PRL 123 208002 (2019), Koumakis et al., Soft Matter 15 7026-7032 (2019), Arlt et al., Nature Communications 10 2321 (2019)), albeit separately.

We thank the reviewer for the comments. We acknowledge their comparison with prior art, but we disagree with their conclusions concerning the novelty of our results. The work by Karani et al. is

highly relevant and we thank the reviewer for bringing it to our attention. It however demonstrates that global behavior emerges from the time modulation of rotational dynamics, without discussing any spatial modulation of D_R , sensorial delay, or the application of a feedback loop on particle dynamics, which are at the core of our findings. The two other works they mention, refer to biological microswimmers with spatial modulations of propulsion velocity, not rotational diffusion. In the manuscript, we already extensively refer to previous work on synthetic and biological microswimmers with position-dependent velocity and emphasize the difference with our strategy. Therefore, we believe that the our manuscript, which now includes a substantial amount of new experimental data, ventures into new exciting directions: In this work, for the first time we use spatio-temporal modulation of rotational diffusion, sensorial delay and feedback strategies to control the behavioral properties of active particles.

Additionally, the most compelling aspect of the work, the localization presented in Figure 4, is shown only in simulations, with no explanation for why this phenomena was (or was not) observed in the experiments. To add further confusion, in the section of the manuscript discussing figure 4 in is not at all clear that the authors have switched to discussing simulations.

We thank the reviewer for the comment. To address the reviewer's concern, we conducted an extensive series of new experiments that shows the emergence of particle localization, in agreement with our simulations. We quantified the degree of localization by measuring the residence time of individual particles in regions of low and high rotational diffusivity at different sampling periods and values of L/v . We find that both the degree of localization and the distribution of residence times are in agreement with the numerical predictions. Both experimental and numerical results are now summarized in the new Figure 5. We also clarified the distinction between numerical and experimental data in relation to the evidence for localization in the main text and in the figures.

Reviewer #2 (Remarks to the Author):

This manuscript investigates the dynamics of active Brownian particles whose rotational diffusivity depends on their position in the 2D-plane. The authors first present an experimental realisation of such a system, consisting of Janus particles that propel in the 2D plane when an AC electric field is applied. In addition, the coated hemisphere of the particles is made to possess a magnetic moment in the direction of propulsion, that aligns with an applied magnetic field. By imposing random variations of the latter combined to a feedback loop, one can alter the dynamics of rotational diffusion, above or below the equilibrium value imposed by the thermal bath. Their feedback loop allows to define a spatially dependent rotational diffusivity, here a chessboard of low and high values, that the particles sense with a given sampling time. They experimentally show that a rich non-Gaussian translational dynamic emerges, which markedly differs from that of active particles with a constant rotational diffusivity, but resembles that of supercooled liquids of interacting passive particles. Using simulations, they show an accumulation (resp. depletion) of particles in regions of high (resp. low) rotational diffusivity, and the sharpness of this pattern can be optimised by varying the sampling time.

I found the manuscript interesting and well organised. The authors demonstrate a nice control of rotational diffusivity above and below its equilibrium value. The experiments and simulations address an important paradigm in collective and transport phenomena, namely how the variation of motility and diffusivity of active particles induce pattern formation. The authors convincingly illustrate that spatial variations of the rotational diffusivity induce pattern formation only if individuals adapt to their environment with some finite delay. They show how non-Gaussian dynamics can emerge as a combination of space-dependent rotational diffusivity and constant self-propulsion, which is not obvious at first sight.

However, I find that the discussions related to Fig. 3 and 4 are unclear and need revision before publication (see my points below). I remain on the fence as for the relevance to a journal like Nature Communications due to the impact of the experimental system. Obtaining good statistics seems prohibitively long in the experiment so many aspects are studied via simulations. Because multiple particles in the field of view would have the same orientations and noise history, it seems to me that the experimental system does not allow for the study of collective effects, which is a drawback. That said, this experimental system may have potential as a probe to drive out of equilibrium various complex materials in a controlled manner.

We thank the reviewer for the positive comments and the thorough review, which has prompted us to perform new experiments and analyses that greatly improve our manuscript. The reviewer is correct in pointing out the limitations of the experimental system in relation to gathering large statistics but also fully mirrors our opinion on the potential of this approach to explore a broad range of dynamical control strategies for active colloidal systems to single out the most interesting ones. Through a new set of experiments we did manage to validate all the key aspects that were previously studied by simulations alone, especially regarding localization. Furthermore, we would like to stress that the full control on the temporal and spatial behavior of our ABPs serves the purpose of demonstrating and “rapidly prototyping” different scenarios. One can then envisage exporting the desired ones, where system-specific feedback and response strategies can be engineered to study collective effects experimentally. We are currently working on a number of these cases.

1. The experimental system is well characterised, and in particular, the control of the rotational diffusivity is well established: the measured distribution of angles corresponds to the imposed one, which testifies that no lag is present in the response of the particle to the field actuation. Could the authors include a short comment on the influence of the field's intensity to achieve such control though?

The maximum strength of the magnetic field in our experimental setup is 65 mT. We find that we can orient our Janus particles along the imposed magnetic field using strengths as low as 10% of the maximum. However, we work with approximately 30-40 mT to ensure that there is no lag in particle reorientation. We have added this detail to the Methods.

2. Fig. 2(e) it takes a very trained eye to evaluate the gaussian/non-gaussian character of the distributions. Would it be possible to plot these in y-log scale instead (gaussians will appear as parabolas and the non-gaussian tails will be clearly visible)? Also, I was wondering what those distributions look like for the rotational degree of freedom.

We thank the reviewer for this good suggestion that prompted us to better characterize the distribution of displacements, also through a new series of experiments aimed at improving statistics. In the new Figure 3, we now plot the distributions on a semi-log plot together with Gaussian and exponential fits for the peak and the tail of the distribution, respectively. A discussion on the emergence of the exponential tail and the similarities with glassy dynamics was also added to the text describing the figure. The distributions of the angular displacements would simply look like the superposition of two Gaussians with different variance, corresponding to the two values of D_R , weighed by the amount of particles in each of the regions.

3. Fig. 2(f) the non-gaussian parameter does not fully decay to zero here but it does in the simulated data. Is it physical or simply due to a lack of statistic in the experiment? Or to the fact that the time

lag does not go beyond 100s, i.e. does not exceed $1/D_R L$?

Thanks to new experiments, the non-Gaussian parameter can now be measured to much longer times with better statistics compared to the original data shown in the old Figure 2f. In the new Figure 3b, we now capture the full decay.

4. The first important result shown here is that non-interacting active Brownian particles with constant propulsion speed exhibit non-gaussian translational dynamics when their rotational diffusivity varies as a function of position. The dynamics resemble passive interacting particles in supercooled liquids but is markedly different from pure active Brownian particles. Nevertheless, I found this section very descriptive of the plots and not always clear. What I am missing is a statement that would summarise and conclude on the influence of the two parameters L/v and τ to help the reader.

We thank the reviewer for the comment and we have now significantly expanded this section, to include more detailed discussions of the emergence of analogies with glassy dynamics, in particular separating the discussion on the effect of L/v and τ .

I would also like the following points to be addressed:

- Page 7, line 135: "For small L/v , the length scale of the spatial variation in D_R is smaller than any persistence length". I guess this regime is achievable in simulations, but not in experiments since the smallest square size is 16 μm , while the smallest persistence length is $v/D_R \tau = 3.5/10 = 0.35\mu\text{m}$. I think I need clarification on why the behaviour is that of a homogeneous rotational diffusivity here.

The sentence above is no longer present in the revised text. This is now discussed extensively in the revised section examining the role of L/v .

- In experiments, is L/v varied through L only or v as well?

We can vary both. For the reported data, we varied L/v by varying L in the range 16-64 μm and v in the range 3-12 $\mu\text{m/s}$. We added this information to the Methods section.

- Page 8, line 153: " α reveals a super-diffusive scaling ..." The different dynamic regimes are easier to see from the MSD than the non-gaussian parameter. The mentioned cut-offs for the sub-diffusive plateau L/v and $3L/v$ seem loosely defined: the yellow band in Fig. 3(d) is clearly much wider than the actual plateau.

We have completely updated this part. The new text and Figure 4 address these issues.

- Page 9, line 159: the peaks at odd multiples of L/v in Fig. 3(c) were not obvious to me on an x-log scale. In all honesty, I thought it was just a lack of statistics at first and would have missed this feature. Could they be magnified in an inset? What would the corresponding MSD look like?

We have added a new figure magnifying this region to the SI (Figure S8).

- Page 9, line 160: "Consequently, the MSDs on timescales L/v decrease and the sub-diffusive scaling gradually vanishes (see Figure 3e). For $\tau > L/v$, α decreases again with no detectable peaks. In this regime, the MSDs increase across all timescales (Fig. 3f)." I am afraid I cannot see a decreasing MSD in Fig. 3(e) and the last statement is a bit empty, since all visible MSDs do increase.

We apologize for the confusion as there was a mistake in the sentence above. In any case, we have completely rewritten this part of the manuscript so the sentence above is not in the revised text.

5. The second important result is that pattern formation can be achieved with a spatially varying rotational diffusivity combined to a non-zero sampling time. Spatial variation of the rotational diffusivity alone does not lead to pattern formation, mainly because it is absent from the flux balance condition. As shown by the authors using simulations, the particle assembly is homogeneous at low and high sampling time. At intermediate values though, particles are localized mainly in regions of high rotational density. The authors discuss the behaviour of the optimum localisation in terms of the sampling times and the crossing time L/v . I like that the authors propose a simple model to estimate the maximum particle fraction that can be localized in high rotational diffusivity regions. But I find the argument unclear and unnecessary long:

- Page 10, line 178: it should be specified that D_R here is D_R^H

- From line 178 still, the penetration time into a ballistic zone is τ but then another time, δt^ , is introduced in line 185. Aren't they the same?*

- The explanation of the model is a bit disorganised and gets heavy in the SI while getting Equation 3 only takes a few steps I find (see additional information in the attached document).

We thank the reviewer for the comment and derivation. Their questions prompted us to rederive the scaling for the localization in a much simpler fashion based on the distribution of residence times in regions of high and low rotational diffusivity. The corresponding new data and analysis is reported in the revised manuscript.

Other minor points:

- Page 2, line 50: "a change of tumbling rate can be straightforwardly mapped into an effective increase of rotational diffusivity". From Ref. 14 it is true that the mapping itself is straightforward but the conditions of its validity seem less so.

Thanks. We have rephrased the sentence.

- Page 2, line 51: "nonetheless, translational and rotational fluctuations are coupled via the thermal bath and their lower bound is dictated by thermal fluctuations". This is only true for active Brownian particles but for run and tumble particles, the tumbling rate is an internal degree of freedom which is not coupled to the bath.

Thanks. We have rephrased also this sentence.

- Page 3: the magnetic moment is mentioned in line 65 but the external control of rotational diffusivity is only described much later in line 73. I think this is a bit confusing for the reader.

Thanks for the comment. We have adapted the text accordingly.

- Page 5, line 101: there is a typo in the word "instantaneous".

Fixed it.

- Page 7, line 127: it may be worth specifying that this non-gaussian parameter is defined for 1D displacements. Why is there a factor $1/5$ in its definition?

We have updated the definition of the non-Gaussian parameter. For clarity we now simply use the excess kurtosis of the distribution of 1D-displacements as a measure of non-Gaussianity throughout the manuscript.

- Page 7, line 131: *I think I would rephrase and say “zero on both short and long timescales”.*

We have rephrased it.

- Page 7, line 139: γ_R is not introduced in the main text.

No longer present in the revised manuscript

- Page 10, line 192: $\langle \eta_t \rangle$ is not introduced in the main text.

No longer present in the revised manuscript

Reviewer #3 (Remarks to the Author):

In their paper the authors demonstrate a quite remarkable control of the rotational diffusion of an active Janus swimmer using a rotating magnetic field. This has never been achieved before and opens the road towards a full control of active swimmers.

The paper is well written, the data and methods are clearly presented and discussed.

And I only have a few minor points to adress (see below)

I therefore recommend publication as soon as these minor points are solved.

We thank the reviewer for the positive feedback.

Minor comments

Fig 2 (e-f) : indicate that the black points are experimental data and that the red lines are numerics

Fixed it in the new Figure 3.

Fig 2 End of caption : one of the D_R^L must be replaced by D_R^H

Fixed it in the new Figure 3.

Just before Formula 1 : replace At equilibrium by In the steady state

Fixed it.

The description of the diffusive regimes is a bit lengthy and lacks some explanations, which only come in the Conclusion and Discussion part. Some reorganization of the writing could easily fix that.

We thank the reviewer for the comment. We now integrated all the explanations in the main text, which was thoroughly revised in terms of structure and clarity.

It would be nice to synthesize the overall findings in a diagram as a function of $\tau L^{-1} v$. A comment about the role of $v/(D_R H L^2)$ would also be welcome.

We thank the reviewer for the comment. The new Figure 5 presents a new rescaling of the localization data as a function of $\tau L^{-1} v$ and $v^2/(D_R H L^2)$. The role of the latter is now also explicitly described in the main text. If the reviewer had a different idea in mind, we would be happy to discuss it.

REVIEWERS' COMMENTS:

Reviewer #1 (Remarks to the Author):

This manuscript has been substantially updated from the previous version, and I now recommend it for publication in this journal. The authors have conducted extensive experiments which now validate the simulation results they have found for localization. These additional efforts have greatly enhanced the manuscript, and emphasized the full range of new behavior that emerges from spatio-temporal control of DR.

Reviewer #2 (Remarks to the Author):

I have carefully read the revised version of the manuscript and the reply letter. The authors have satisfactorily addressed all my comments and even went beyond expectations. Indeed, the addition of new experimental data and the extensive revision of the discussion bring this work to a whole new level and is incomparably better than the previous version. The discussion of the non-Gaussian features on experimental data is now very clear. I also appreciate the use of residence times in the localisation part, which improves clarity and allows for a direct comparison between experiments and simulations. Finally, the addition of Fig 6 shows that a system can be more stable when agents are autonomous than when they obey to a centralised control, which is an important point in my opinion. This work is an important contribution in the field of programmable active matter and also for the fundamental understanding of spatial organisation. I therefore recommend publication in Nature Communications in its current form.

Very minor comments:

- Line 192: use lowercase v
- Line 220 and 264: not clear why penetration is by $2v\tau$ and not $v\tau$.
- Line 275: keep the order H/L as for the rest of the paragraph to improve clarity: "TH and TL scale as $(L/v)^2$ and L/v respectively".

Reviewer #3 (Remarks to the Author):

In this revised version of their paper, the authors have on one hand significantly increased the number of experimental and numerical analysis, on the other hand clarified and singled out the respective role of the two main control parameters, namely the sensorial delay τ and the ballistic time L/v .

Also the figures are much better presented and discussed than in the previous version.

The present work is not only an experimental tour de force, it also provide significant insight on how smart control over active systems can be achieved.

I therefore recommend publication in the present form of the manuscript.

We thank all three referees for recommending publication and for the appreciation of our additional work to improve our manuscript following their initial comments. We address all remaining concerns in the document below. The original reviewers' comments are in italic.

REVIEWERS' COMMENTS:

Reviewer #1 (Remarks to the Author):

This manuscript has been substantially updated from the previous version, and I now recommend it for publication in this journal. The authors have conducted extensive experiments which now validate the simulation results they have found for localization. These additional efforts have greatly enhanced the manuscript, and emphasized the full range of new behavior that emerges from spatio-temporal control of DR.

We thank the referee for the positive comments.

Reviewer #2 (Remarks to the Author):

I have carefully read the revised version of the manuscript and the reply letter. The authors have satisfactorily addressed all my comments and even went beyond expectations. Indeed, the addition of new experimental data and the extensive revision of the discussion bring this work to a whole new level and is incomparably better than the previous version. The discussion of the non-Gaussian features on experimental data is now very clear. I also appreciate the use of residence times in the localisation part, which improves clarity and allows for a direct comparison between experiments and simulations. Finally, the addition of Fig 6 shows that a system can be more stable when agents are autonomous than when they obey to a centralised control, which is an important point in my opinion.

This work is an important contribution in the field of programmable active matter and also for the fundamental understanding of spatial organisation. I therefore recommend publication in Nature Communications in its current form.

We thank the reviewer for the careful reading and the positive feedback.

Very minor comments:

- Line 192: use lowercase v

Thank you. This has been fixed.

- Line 220 and 264: not clear why penetration is by $2v\tau$ and not $v\tau$.

This is due to the fact that the sampling takes place every τ and the value of D_R at time $n\tau$ is updated based on the position at time $(n-1)\tau$, as it is shown in Figure 2a. We have added a pointer to Figure 2a in line 220 (now line 223) that shows it clearly, and rewritten the penetration length as $v(2\tau)$ to make it even more apparent.

- Line 275: keep the order H/L as for the rest of the paragraph to improve clarity: "TH and TL scale as $(L/v)^2$ and L/v respectively".

Thank you. We have updated the order following the suggestion.

Reviewer #3 (Remarks to the Author):

In this revised version of their paper, the authors have on one hand significantly increased the number of experimental and numerical analysis, on the other hand clarified and singled out the respective role of the two main control parameters, namely the sensorial delay τ and the ballistic time L/v .

Also the figures are much better presented and discussed than in the previous version.

The present work is not only an experimental tour de force, it also provide significant insight on how smart control over active systems can be achieved.

I therefore recommend publication in the present form of the manuscript.

We thank the reviewer for the positive feedback.